# Tuberculous Pleural Effusion-Derived Exosomal miR-130b-3p and miR-423-5p Promote the Proliferation of Lung Cancer Cells via Cyclin D1

**DOI:** 10.3390/ijms251810119

**Published:** 2024-09-20

**Authors:** Hyun-Jung Kang, Sangho Yun, Seung-Ho Shin, Dong Hyuk Youn, Ga-Hyun Son, Jae Jun Lee, Ji Young Hong

**Affiliations:** 1Institute of New Frontier Research Team, Hallym University College of Medicine, Chuncheon 24252, Republic of Korea; kangsang8106@naver.com (H.-J.K.); z4213@daum.net (S.Y.); sin12ho@gmail.com (S.-H.S.); zk61326@naver.com (D.H.Y.); ntr5017@naver.com (G.-H.S.); 2Division of Pulmonary and Critical Care Medicine, Department of Medicine, Chuncheon Sacred Heart Hospital, Hallym University Medical Center, Chuncheon 24252, Republic of Korea

**Keywords:** tuberculosis, lung cancer, exosome, intercellular communication

## Abstract

Epidemiologic studies have shown an association between tuberculosis and lung cancer. The altered tumor microenvironment after tuberculosis infection appears to contribute to cancer progression. Pleural effusions are enriched in exosomes, which act as mediators of intercellular communication. We hypothesized that tuberculous pleural effusion (TPE)-derived exosomes mediate intercellular communication. Then, we examined the interaction between TPE-derived exosomes and cancer cells. Exosomal miRNA profiling of TPE was performed using a microRNA array. An in vitro lung cancer cell experiment and an in vivo mouse xenograft tumor model were used to evaluate the effects of the selected exosomal microRNAs. TPE-derived exosome treatment enhanced the growth of A549 cells both in vitro and in a nude mouse xenograft model. Neighboring cancer cells were observed to take up TPE-derived exosomes, which promoted cancer cell invasion. Exosome-mediated transfer of the selected microRNAs, including miR-130b-3p and miR-423-5p, to A549 lung cancer cells activated cyclin D1 signaling and increased the expression of phosphorylated p65, a cyclin D1 transcription factor. Inhibitors of miR-130b and miR-423-5p suppressed the promotion of lung cancer by TPE-derived exosomes and reduced the expression of p65 and cyclin D1. These results suggest that TPE-derived exosomal miRNAs can serve as a novel therapeutic target in tuberculous fibrosis-induced lung cancer.

## 1. Introduction

Pleural effusion (PE), an abnormal collection of fluid in the pleural space, is a common manifestation of many diseases. In clinical practice, PE analysis is frequently used for diagnostic purposes.

Exosomes are phospholipid bilayer vesicles that contain protein, lipids, mRNA, and microRNAs [1]. They are present in high concentrations in bodily fluids, including blood, urine, bronchoalveolar lavage, and PE [2]. PE-derived exosomes are an ideal source of biomarkers in lung cancer, mesothelioma, and tuberculosis (TB) [3,4,5]. They mediate intercellular communication by transmitting biological signals to recipient cells [6]. Recent studies have shown that exosomal miRNAs influence mRNA translation by immune cells, thereby contributing to cancer progression via immunoregulatory and angiogenic effects [7,8].

Both TB and lung cancer remain significant public health concerns and are leading causes of death worldwide [9,10]. Epidemiological studies have reported an association between TB and the risk of lung cancer [11,12]. In addition, prior TB correlates with increased mortality from lung cancer [13]. Although the precise mechanisms underlying these relationships are not fully understood, the altered lung microenvironment following TB may contribute to the development or progression of lung cancer [14,15,16]. In an experimental study, macrophages injected with Mycobacterium tuberculosis produced epiregulin, which is implicated in squamous metaplasia and tumorigenesis in the lung [14]. Woo et al. found that tuberculous fibrosis enhances tumorigenic potential via the NOX4–autophagy axis [15]. In previous research, we demonstrated that Arg 1+ M2 polarization of macrophages induced by tuberculous PE mediates lung cancer proliferation by increasing autophagy signaling and E-cadherin expression [16]. Then, we hypothesized that exosomal miRNAs derived from tuberculous pleural effusion (TPE) play a role in lung cancer progression, which led us to investigate intercellular communication between TPE-derived exosomal miRNAs and lung cancer. Specifically, we conducted in vitro experiments to evaluate the potential of exosomal miRNAs as therapeutic targets for post-TB lung cancer.

## 2. Results

### 2.1. TPE-Derived Exosomes Induce the Invasion of Lung Cancer Cells

Exosomes from TPE and transudative PE were isolated using the Exo2D EV isolation kit. Their diameters were determined by TEM and ranged from 22.2 to 40.5 nm. TPE-derived exosomes were significantly larger than transudate-derived exosomes (31.1 ± 4.7 nm vs. 28.1 ± 3.6 nm, *p* = 0.04, Figure 1). Expression of the exosomal markers CD63, CD9, and CD81 was confirmed by Western blotting.

Prior to the experiments investigating the effect of TPE-derived exosomes on the malignant potential of lung cancer, a cellular uptake assessment was performed to evaluate exosome internalization by A549 cells. PKH26 was used as a cell tracer by binding to the cell membrane of A549 cells. A549 cells and TPE-derived exosomes were stained with PKH26 (red) and PKH67 (green) dyes, respectively. Thus, PKH67-labeled TPE-derived exosomes were incubated with PKH26-labeled A549 cells for 3 h, after which the fluorescence intensity was measured using flow cytometry. Fluorescence microscopy confirmed the uptake of PKH67-labeled exosomes by A549 cells, based on the colocalization of red fluorescence and green fluorescent exosomes (Figure 1). According to the flow cytometry analysis, 81.4% of A549 cells were exosome-positive after treatment.

Then, the role of TPE-derived exosomes in the migration and invasion of lung cancer cells was analyzed (Figure 1). In the wound healing assay, the migration of A549 cells was significantly enhanced in the presence of TPE-derived exosomes compared to untreated cells. In the Transwell assay, TPE-derived exosomes promoted the migration and invasion of lung cancer cells compared to untreated cells. Transudate-derived exosomes induced lung cancer invasion but to a lesser extent than TPE-derived exosomes.

A xenograft model was established to examine the effect of TPE-derived exosomes on lung cancer cells (Figure 1). Macroscopically, A549 xenografts treated intratumorally with TPE-derived exosomes showed faster tumor growth and appeared more vascularized than the PBS-treated xenografts.

### 2.2. Identifying Exosomal miRNAs from TPE Involved in the Invasion of Lung Cancer Cells

Next, the miRNA expression profile of TPE-derived exosomes was examined. Total RNA was extracted and subjected to miRNA microarray analysis. In comparisons with cells treated with transudate-derived exosomes, >2-fold increases in the exosomal miRNAs miR-130b-3p, miR-423-5p, miR-142-5p, miR-107, miR-320b, and miR-185-5p, and >2-fold decreases in miR-374a-5p, miR-409-3p, miR-125b-5p, miR-224-5p, and miR 139-5p were observed in cells treated with TPE-derived exosomes (Figure 2). A subset of these miRNA candidates from the in vitro and in vivo experiments were verified by qPCR. The levels of miR-130b-3p and miR-423-5p were significantly higher in cancer cells treated with TPE-derived exosomes than in those treated with transudate-derived exosomes.

Next, nude mice were subcutaneously injected with A549 cells, and the resulting tumors were injected intratumorally with TPE-derived exosomes seven times when the mean tumor volume reached 30 mm^3^. The levels of miR-130b-3p and miR-423-5p in the xenograft tissues treated with TPE-derived exosomes were higher than those in the PBS-treated group (Figure 2). Based on the similar expression patterns in vitro and in vivo, miR-130b-3p and miR-423-5p were selected for further investigation. 

### 2.3. GW4869 Inhibits the Migration and Invasion of TPE-Derived Exosome-Stimulated A549 Cells

In a wound healing assay at 24 and 48 h, the mobility of A549 cells treated with TPE exosomes and GW4869 (24 h: 23.70 ± 2.79%, 48 h: 44.74 ± 3.93%) was significantly lower than that of cells treated with TPE exosomes alone (24 h: 39.60 ± 3.34%, 48 h: 64.05 ± 5.51%) (both *p* < 0.05, Figure 3A). In the Transwell invasion assay, combined TPE exosome and GW4869 treatment significantly inhibited A549 cell invasion and migration, by 112.4%, compared to TPE treatment alone (329.9 ± 36.95 vs. 217.5 ± 16.43, *p* < 0.05, Figure 3B). The effect of GW4869 in cells treated with transudate-derived exosomes was less pronounced than with TPE-derived exosomes (237.3 ± 6.14 vs. 199.8 ± 20.15, *p* = 0.149). The expression of *p*-p65, p65, and cyclin D in A549 cells was examined by Western blotting, which showed higher levels of *p*-p65, p65, and cyclin D in cells treated with TPE-derived exosomes than in the control group (Figure 3C). However, these increases were reduced by combined TPE exosome and GW4869 treatment. Taken together, these results indicate that TPE exosomes promote lung cancer progression through the p65-cyclin D pathway. The exosome release inhibitor GW4869 blocked this effect.

### 2.4. Exosome-Mediated Transfer miR-130 and miR-423 from TPE Promotes the Invasion of Cancer Cells via Cyclin D

The contribution of the transfer of select miRNAs from TPE-derived exosomes to lung cancer cell invasion was investigated by transfecting A549 cells with inhibitors of miR-130b-3p and miR-423-5p before their treatment with TPE-derived exosomes. In both the wound healing and Transwell invasion assays, the two inhibitors reversed the enhanced migration and invasion induced by TPE-derived exosomes (Figure 4). Based on our data showing that TPE regulates cell cycle proliferation through the p65–cyclin D pathway, the effect of exosomal inhibitors on regulators of the cell cycle was examined. Cyclin A, cyclin D1, and p21 are regulators of the G1 to S phase transition in mammals [17]. The levels of *p*-p65 and cyclin D1 were lower and those pf P21 were higher in A549 cells transfected with miR-130b-3p and miR-423-5p inhibitors before their treatment with TPE-derived exosomes than in cells treated with TPE-derived exosomes alone. The level of cyclin A was not significantly affected.

## 3. Discussion

miRNAs, small non-coding endogenous molecules, are powerful regulators of mRNA expression and translation [18]. Exosomes are mediators of intercellular communication that contain higher proportions of miRNA than in donor cells [19,20]. Exosomal miRNAs can affect tumor growth, invasion, metastasis, angiogenesis, and drug resistance [21]. As body fluids are enriched in exosomes, exosome-based liquid biopsy is emerging as a promising method for the diagnosis and prognosis of a wide range of diseases [22,23].

The association between TB and lung cancer has been reported in clinical and basic research studies [24,25]. Pleural mesothelial cells contribute to PE formation by expressing intercellular molecules and inducing cell migration and protein leakage [26]. In pleural mesothelial cells treated with heat-killed M. tuberculosis, both cell migration and the invasion of lung cancer cells via NOX4 signaling have been demonstrated [15]. These findings suggest that TPE-derived exosomes promote lung cancer progression. Our data from both in vivo and in vitro studies confirmed that TPE-derived exosomes promote lung cancer proliferation, and this effect was reduced when GW4860, an inhibitor of exosome biogenesis, was administered. We focused on the miRNA component of the exosomes.

Our study revealed that the expression of several miRNAs differs between TPE exosomes and transudate-derived exosomes. Notably, the PCR sequencing results from nude mice experiments and cell studies showed that miR-130b-3p and miR-423-5p were significantly overexpressed in A549 cells treated with tuberculous exosomes. Exosomes mediate their effects on recipient cells through a multi-step process: interaction with cell surface receptors, internalization via endocytosis, and the release of the endosome’s contents into the cytoplasm, where they carry out their functional effects [27]. After exosomes are delivered to recipient cells, some of the miRNAs they contain function within the cell, while others undergo degradation. This difference is determined by factors such as miRNA stability, the intracellular environment, interactions with target mRNAs, and the metabolic state of the cell. The delivery of specific miRNAs reflects the interplay between exosomal surface proteins and ligands, receptor specificity, environmental factors, host genetics, and the structural characteristics and stability of the miRNA [28,29]. 

Previous studies of the roles of miR-130b-3p and miR-423-5p in cancer progression have shown that miR-130b promotes cell growth and chemo-resistance by regulating the PTEN gene in lung cancer, renal cell carcinoma, and breast cancer [30,31,32], whereas miR-130b inhibits tumor migration and invasion in prostate cancer by suppressing the DLL1 and PI3K/Akt pathways [33]. The reduced expression of miR-130b is associated with a diagnosis of prostate cancer but with a good prognosis in colorectal cancer [34,35]. Similarly, miR-423-5p can have oncogenic or tumor suppression effects depending on the cancer type. For example, miR-423-5p induces the expression of caspase 3, caspase 9, and p53 as a tumor suppressor in colon cancer [36] but aggravates lung adenocarcinoma via the downregulation of CADM1 expression [37]. 

Several studies have demonstrated that assessments of a combination of exosomal miRNAs are of higher diagnostic power than a single unique miRNA, due to the overlap of target genes [38,39]. TPE-derived exosomes affect pathways that regulate the G1 to S transition in lung cancer, evidenced by an increase in cyclin D1 and a decrease in p21. Cyclin D1 is a cell cycle regulatory protein that activates CDK4/6 while p21 is a CDK inhibitor. The overexpression of cyclin D1 has oncogenic consequences due to the activation of the RAS–MEK–ERK and PI3K pathways. One of the mechanisms regulating cyclin D overexpression is a reduction in the levels of the miRNAs targeting CCND1 [40]. The downregulation of miR-15a and miR-16 clusters has been shown to correlate inversely with cyclin D levels in both lung and prostate cancer [41,42]. In contrast to the miR-15a and miR-16 clusters, which directly target and inhibit CCND1, miR-130b and miR-423 indirectly activate cyclin D1 via NF-κB signaling. 

The human cyclin D1 promoter contains two putative NF-kB binding sites, termed D1-kB1 and D1-kB2. NF-κB contributes to cell cycle progression by binding to the cyclin D1 promotor and activating transcription [43]. 

Our results showed that inhibitors of miR-130 and miR-423 decreased NF-kB levels in A549 cells following their increase in response to TB exosomes. This is consistent with previous research showing that miR-130 and miR-423 increase NF-kB levels [44,45]. In breast cancer, miR-423 activates the NF-κB pathway by regulating TNIP2 [44]. In bladder cancer, miR-130b contributes to its malignant potential by inducing the persistent activation of NF-κB through a positive feedback mechanism [45].

We also found that a combination of miR-130b-3p and miR-423-5p inhibitors not only inhibited cyclin D1 but also upregulated p21, leading to cell cycle arrest and the inhibition of lung cancer growth. Both miR-130 and miR-423 bind to and inhibit the p21 promoter. Brock et al. reported that hypoxia-induced miR-130 directly targets the tumor suppressor p21 in vitro and in vivo [46]. In hepatocellular carcinoma cells, miR-423 promotes cell cycle progression at the G1/S transition by targeting p21Cip1/Waf1 [47]. In summary, TPE-derived exosomal miR-130 and miR-423 influence cell cycle regulation by inducing an increase in cyclin D1 levels via NF-kB and by directly inhibiting p21, thereby contributing to cancer proliferation. As a limitation, in addition to the NF-κB pathway demonstrated in this study, further mechanistic research should be conducted to explore whether these miRNAs also influence cyclin D1 by regulating other pathways.

This study thus provides novel insights into the association between TB and lung cancer. TPE-derived exosomal miR-130 and miR-423 were identified by a microRNA array. Their exosome-mediated transfer in lung cancer was shown to enhance the malignant potential of A549 cells by activating cell cycle progression. From the perspective of personalized medicine, developing therapeutic targets or novel biomarkers for tuberculosis-related lung cancer requires a thorough understanding of the precise regulatory mechanisms of EV miRNA transfer and their biological functions to identify target genes. A key challenge in adapting miRNA inhibitors for therapy is increasing their delivery efficiency. Additionally, establishing a large-scale tuberculosis-related lung cancer cohort is essential for validating candidate miRNAs and target genes.

## 4. Materials and Methods

### 4.1. Patients and Samples

Pleural effusion samples were collected from patients at Chuncheon Sacred Heart Hospital during routine thoracentesis for diagnostic purposes. All human experiments were performed in accordance with the Declaration of Helsinki. The collection and use of these samples were approved by the hospital’s Research Ethics Committee (Institutional Review Board number: 2012-27). The patients consented to sample collection and the disclosure of their anonymized information.

PE (10 mL) was collected into sterile tubes from all patients via thoracentesis. The samples were centrifuged at 3000× *g* for 10 min and supernatants were frozen at −80 °C for analysis. The samples were divided into exudative or transudative PEs according to Light’s criteria [48]. Among the exudative PEs, tuberculous effusion was defined as pleural fluid that was culture-positive for M. tuberculosis or from patients with a tuberculous infection histologically confirmed by pleural biopsy. The transudative PEs were used as a reference control group. The clinical characteristics of the patients are listed in Table 1. We compared exudative tuberculous pleural effusion-derived exosomes (TPE-derived exosomes) with transudative pleural effusion-derived exosomes (T-exosomes).

### 4.2. Exosome Isolation and Quantification

Exosomes were isolated using an Exo2d-EV isolation kit (Exosome Plus, Inc., Seoul, Republic of Korea) according to the manufacturer’s instructions. Briefly, 5 mL PE was centrifuged at 3000× *g* for 15 min to remove cell debris and larger vesicles. From the supernatant, 250 µL was mixed with 50 µL Exo2D solution and incubated for 30 min at 4 °C. After centrifugation at 3000× *g* for 30 min, the exosome pellet was resuspended in 100 µL phosphate-buffered saline for further analysis. Total RNA, extracted from the purified exosomes using TRIzol reagent (Thermo Fisher Scientific, Waltham, MA, USA) as per the standard protocol, served as the basis for the RNA analysis and sequencing. The protein concentration in the exosomes was measured before the experiment using the BCA protein assay (Thermo Fisher Scientific, Waltham, MA, USA).

### 4.3. Transmission Electron Microscopy 

A549 cells treated with TPE-derived exosomes or transudate-derived exosomes for 48 h were examined by field emission transmission electron microscopy (FE-TEM, JEM 2100F; JEOL, Tokyo, Japan).

### 4.4. Small RNA Library Construction and miRNA Sequencing

A cDNA library was constructed with 10 ng input RNA, using a SMARTer smRNA-seq kit (Illumina, San Diego, CA, USA) and Illumina’s standard procedure. Briefly, RNA fractions were selected, adapters were ligated onto the RNAs, and then these were amplified to construct a library suitable for high-throughput sequencing. The obtained smRNA library was sequenced on an Illumina Hiseq 2500 genome analyzer platform. Raw sequence reads were refined in a filtering process and after the trimming of adapter sequences. The resulting high-quality, processed reads were sequentially aligned with reference genomes for classification and analysis. Known miRNAs were identified using miRbase v22.1 and novel miRNAs were predicted using miRDeep2 RNAcentral 14.0. Non-coding RNA was used to classify other types of RNA sequences. Differentially expressed miRNAs were identified using statistical methods, including fold-change calculation and the exactTest function from edgeR (version 3.9) and hierarchical clustering. All procedures for smRNA library construction, miRNA sequencing, and data analysis were conducted by Macrogen Inc. (Seoul, Republic of Korea).

### 4.5. Animal Experiments

Male BALB/c nude mice (*n* = 12, age, 7 weeks; weight, 18 ± 2 g) were purchased from DooYeol Biotech (Seoul, Republic of Korea). All animals were housed at 21–23 °C and 51–54% humidity in a pathogen-free environment on a 12/12 h light/dark cycle and allowed free access to food and water. The mice were monitored daily for health and behavior. For xenograft generation, 2 × 10^6^ A549 cells were subcutaneously transplanted into the flanks of 12 mice. Then, the mice were maintained according to the United Kingdom Coordinating Committee on Cancer Research guidelines. Tumor volumes, calculated as (tumor length × width^2^)/2, were monitored every other day through caliper measurement [49]. Seven days after transplantation, when the tumors had reached a volume of 30 mm^3^, tuberculous effusion-derived exosomes or transudative pleural effusion-derived exosomes were injected intratumorally at a dose of 100 μg per mouse every two days seven times. On day 21, the tumor masses were measured and the tumors were excised for analysis. qRT-PCR was performed on xenograft RNA to determine the effect of tuberculous effusion-derived exosomes on A549 cells. This experiment was carried out in duplicate. All animal experiments were reviewed and approved by the Institutional Animal Care and Use Committee of Hallym University (no: Hallym 2017–47).

### 4.6. Cell Culture and Reagent Treatment

The human adenocarcinoma cell line A549 was purchased from the American Type Culture Collection (ATCC, Manassas, VA, USA). The cells were cultured according to the manufacturer’s instructions and maintained in RPMI 1640 medium (BYLABS, Hanam, Republic of Korea) supplemented with 10% fetal bovine serum (FBS), 100 U penicillin/mL, and 100 μg streptomycin/mL at 37 °C in a humidified 5% CO^2^ atmosphere incubator.

GW4869, an exosome inhibitor, was obtained from Millipore Sigma (Burlington, MA, USA) and dissolved in DMSO for use in the experiment. TPE-derived exosomes or transudate-derived exosomes were diluted in PBS and used in the A549 cell experiments at a dose of 50 μg/mL.

### 4.7. Cellular Internalization of Exosomes

TPE-derived exosome/cancer cell interactions were assessed in vitro by treating 1 × 10^6^ PKH26-labeled A549 cells with 50 µg PKH67-labeled TPE-derived exosomes or a PKH67-PBS control. After a 3 h incubation at 37 °C, the uptake of fluorescently labeled exosomes by the cells was examined via fluorescence microscopy and flow cytometry.

### 4.8. Cell Migration and Invasion Assays

A scratch wound healing assay was performed to compare the effect of TPE-derived and transudate-derived exosomes on the migration of A549 cells. The cells (5 × 10^5^) were seeded onto a 6-well cell culture plate containing RPMI culture medium supplemented with 4% FBS and incubated at 37 °C in a 5% CO_2_ incubator. When the cells reached 95–100% confluence, a scratch wound was made in the monolayer using a sterile pipette tip. The cells were allowed to grow as described above for an additional 24 and 48 h in the presence of TPE-derived or transudate-derived exosomes. Images (magnification: ×100) were taken at 0, 24, and 48 h using an inverted light microscope (Olympus Corp., Tokyo, Japan). The migration distance was quantified using ImageJ software (Version 1.54k, National Institutes of Health, Bethesda, MD, USA; https://imagej.net/ij/ (accessed on 1 September 2024). The percentage of area closure was calculated as the final wound width/initial wound width × 100.

In addition, an A549 cell invasion assay was performed using Transwell chambers with an 8 µm pore size (Corning, Glendale, AZ, USA). Diluted Matrigel (50 µL; 3 mg protein/mL; final concentration: 200 μg/mL) was added to the center of each well in the upper chamber. Then, the chambers were incubated at room temperature for 1 h to allow them to be coated. A549 cells treated (or not) with GW4869 or miRNA inhibitors were seeded in the upper chamber, and TPE-derived or transudate-derived exosomes were seeded in the lower chamber. After 24 h, the invasive cells that had penetrated the Matrigel barrier at the lower surface were stained with crystal violet solution. The bound crystal violet was eluted by adding 400 µL 33% acetic acid into each insert followed by shaking for 10 min. The eluate from the lower chamber was transferred to a 96-well microplate and the absorbance at 590 nm was measured using a plate reader.

### 4.9. Extraction of miRNA, Real-Time PCR, and Western Blotting

miRNA was extracted from A549 or nude mouse cancer tissue using the miRNeasy mini kit (Qiagen, Hilden, Germany) according to the manufacturer’s protocol and then quantified by measuring the optical density at 260 nm. cDNA was synthesized using the Mir-X TM miRNA first-strand synthesis kit (Clontech Laboratories, Inc., Mountain View, CA, USA). Real-time PCR was performed on a Rotor-Gene Q system, using SYBR Green I as a double-stranded-DNA-specific dye, according to the manufacturer’s instructions. The cDNAs were PCR-amplified as follows: 40 cycles of denaturation at 94 °C for 15 s, annealing at 58 °C for 30 s, and extension at 72 °C for 30 s. Relative expression was calculated as ddCt, and the data were normalized by comparison with the expression of β-actin in the control group. The primer sequences used for each gene are listed in Appendix A.

For Western blotting analysis, A549 cells were lysed with RIPA buffer containing a protease inhibitor cocktail (GenDEPOT, Baker, TX, USA). The membranes containing the transferred proteins were incubated overnight at 4 °C with the following primary antibodies: β-actin, CD63, and CD81 (Abbkine, Atlanta, Georgia, USA); CD9 (LifeSpan BioSciences, Shirley, MA, USA); phosphorylated p65 (*p*-p65), p65, cyclin D1, cyclin A, and p21 (Cell Signaling Technology, Danvers, MA, USA). The blots were washed in PBS with Tween-20 and then incubated with the corresponding horseradish-peroxidase-conjugated secondary antibodies. Signals were enhanced and detected using ECL detection (Thermo Fisher Scientific, Waltham, MA, USA).

### 4.10. Transfection of miRNA Inhibitors

To determine the role of cyclin D1 in the proliferation of lung cancer cells exposed to TPE-derived exosomes, A549 cells were treated with inhibitors for human miR-130b-3p and miR-423-5p (200 pM; BioNEER, Daejeon, Republic of Korea). The cells were transfected using the in vitro siRNA/miRNA transfection reagent INTERFERin (Polyplus, New York, NY, USA), according to the manufacturer’s protocol. Transfection efficiency was determined by Western blotting.

### 4.11. Statistical Analysis

The data are presented as the mean ± SEM and were analyzed using a *t*-test or two-way ANOVA (Prism v8.02; GraphPad Software Inc., San Diego, CA, USA) followed by correction for multiple comparisons. The results were considered significant at a *p* value < 0.05.

## Figures and Tables

**Figure 1 ijms-25-10119-f001:**
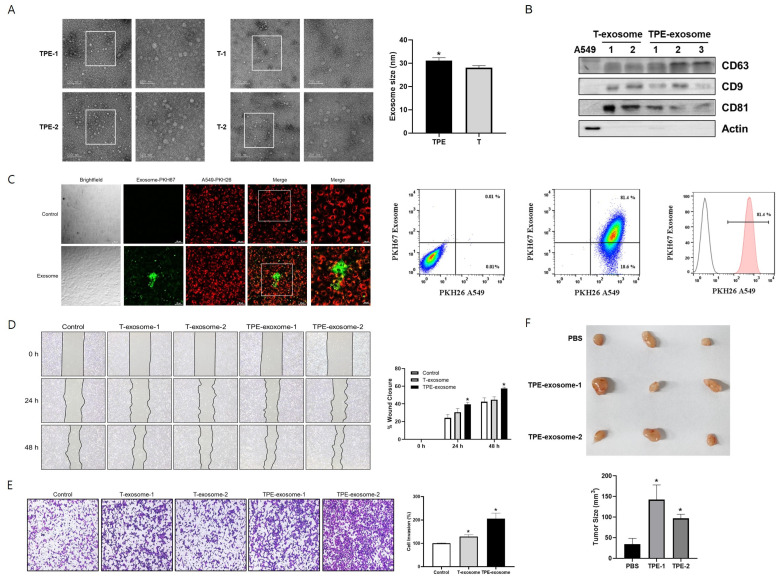
TPE-derived exosomes promote the migration and proliferation of lung cancer cells. Exosomes derived from tuberculous pleural effusions (TPEs) were isolated by ultracentrifugation as described in the Methods section. (**A**) Transmission electron microscopy images of isolated exosomes. Scale bars = 200 nm (left column) and 100 nm (right column). * *p* < 0.05 (**B**) The expressions of exosomal markers in TPE-derived exosomes, as detected on Western blots. (**C**) Assessment of TPE-derived exosome/cancer cell interactions in vitro. Samples (50 µg) of PKH67-labeled TPE-derived exosomes or a PKH67-PBS control were added to 1 × 10^6^ PKH26-labeled A549 cells and incubated at 37 °C for 3 h. The uptake of fluorescently labeled exosomes by A549 was detected by fluorescence microscopy and flow cytometry, both of which strongly suggested a high binding affinity of TPE-derived exosomes for A549 cells. (**D**,**E**) Cell migration and invasion as measured in scratch wound healing and Transwell assays. * *p* < 0.05 vs. control (**F**) Athymic nude mice were subcutaneously injected with A549 cells. When the mean tumor volume reached 30 mm^3^, the mice were intratumorally injected with TPE-derived exosomes or PBS. Representative photographs of tumors excised from each experimental group.

**Figure 2 ijms-25-10119-f002:**
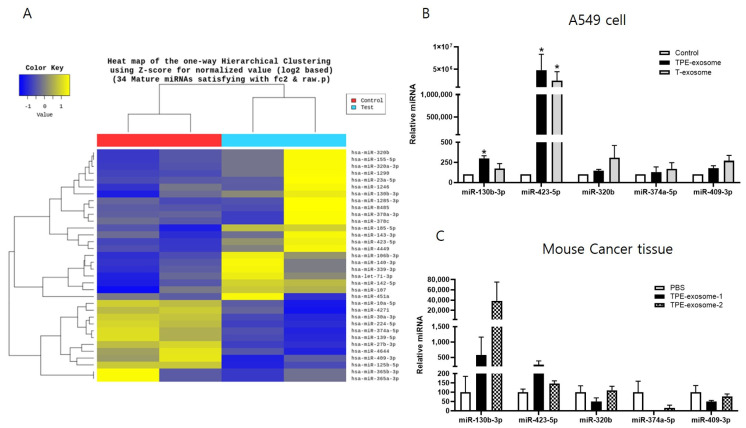
Identification of exosomal miRNAs transferred from TPE–derived exosomes to A549 cancer cells. (**A**) Hierarchical clustering analysis of exosomal microRNA expression. A clustering analysis was performed on exosomal microRNAs differentially expressed between TPE–derived and transudate–derived exosomes. Columns represent individual samples, and rows show each exosomal microRNA. Yellow and blue in the cells indicate high and low expression levels, respectively, according to the scale bar (log2–transformed scale). (**B**) The expression of TPE–derived exosomal miRNAs in cancer cells, as measured via quantitative real–time PCR. A549 cells were treated with TPE–derived or transudate-derived exosomes. * *p* < 0.05 vs. control (**C**) Expression of selected exosomal miRNAs from the xenograft tissues of mice injected intratumorally with TPE–derived exosomes. All experiments were performed in triplicate.

**Figure 3 ijms-25-10119-f003:**
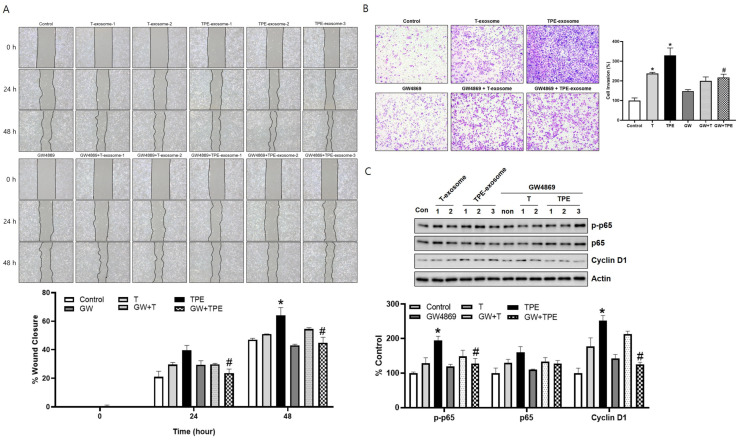
GW4869 inhibits the cellular migration and invasion induced by TPE-derived exosomes. (**A**) The migration rates of A549 cells exposed to TPE-derived or transudate-derived exosomes and with or without GW4869 treatment were measured in wound healing assays (magnification: ×100). * *p* < 0.05 vs. control, # *p* < 0.05 vs. TPE. (**B**) The invasion rates of A549 cells exposed to TPE-derived or transudate-derived exosomes and with or without GW4869 treatment were measured in Transwell assays (magnification: ×100). * *p* < 0.05 vs. control, # *p* < 0.05 vs. TPE. (**C**) Expression of p65 and cyclin D1 as measured via Western blotting in A549 cells treated with TPE-derived or transudate-derived exosomes with or without GW4869. All experiments were performed in triplicate.

**Figure 4 ijms-25-10119-f004:**
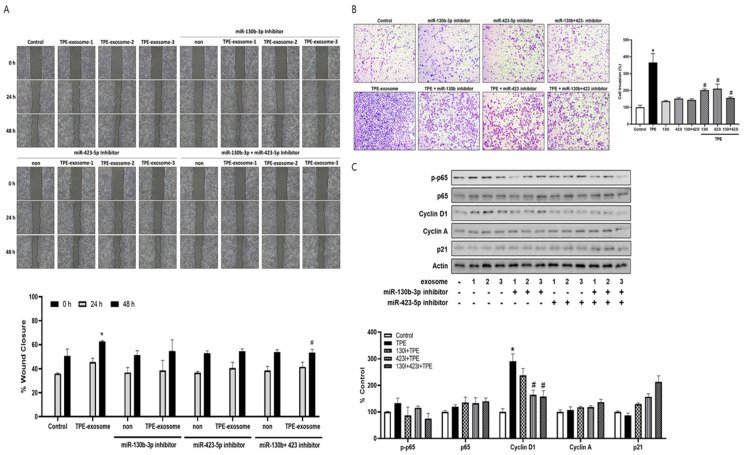
Exosome-mediated transfer of miR-130b-3p and miR-423-5p from TPE promotes the invasion of cancer cells via cyclin D1. A549 cells were treated with TPE-derived exosomes followed by transfection with inhibitors of miR-130b-3p and miR-423-5p. An equal volume of PBS served as a blank control. (**A**) Representative micrographs of the wound healing assay (magnification: ×100). The average migration distance was calculated based on the difference in the gap widths of the same area. * *p* < 0.05 vs. time control, # *p* < 0.05 vs. TPE-derived exosomes (**B**) Representative micrographs of the Transwell assay (magnification: ×100). The number of invasive cells per high-power field was calculated based on three independent experiments. * *p* < 0.05 vs. time control, # *p* < 0.05 vs. TPE-derived exosomes (**C**) Expression of p65 and G1/S regulatory molecules as measured via Western blotting in A549 cells treated with TPE-derived exosomes with or without inhibitors of miR-130b-3p and miR-423-5p. All experiments were performed in triplicate. The results are expressed as the mean ± SEM of triplicates.

**Table 1 ijms-25-10119-t001:** Clinical characteristics of the study participants.

Characteristics	TPE1	TPE2	T1	T2
Age	23	77	51	51
Sex	Male	Male	Male	Male
Comorbidity	None	HTN, CAOD	Cirrhosis, CKD	Wernicke’s encephalopathy
Pleural fluid analysis				
Leukocyte (/mm^3^)	288	1530	576	46
Total protein (g/dL)	5.4	4.7	3	<2.0
Albumin	3.6	2.5	1.8	1.2
LDH (IU/L)	1613	178	76	61
CRP (mg/L)	6.7	6.2	<4.0	4.8
ADA (IU/L)	102.5	77.5	29	6
Neutrophil (%)	20	20	60	5
Lymphocyte (%)	80	80	40	95
Serum				
Total protein (g/dL)	7.1	6.4	6.5	5.2
Albumin (g/dL)	4.6	3.4	3.6	3.1
LDH (IU/L)	225	229	176	177
Light criteria				
Pleural protein/serum protein > 0.5	Yes	Yes	No	No
Pleural LDH/serum LDH > 0.6	Yes	Yes	No	No
Pleural LDH > 2/3 of serum LDH	Yes	Yes	No	No
Positive pleural fluid TB culture	Yes	Yes	No	No

TPE, tuberculous pleural effusion; T, transudative effusion; HTN, hypertension; CAOD, coronary artery obstructive disease; CKD: chronic kidney disease; LDH, lactate dehydrogenase; CRP, C-reactive protein; ADA, adenosine deaminase; TB, tuberculosis.

## Data Availability

The data used to support this research are available from the corresponding author upon request.

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
