# Peer review of "Tuberculous Pleural Effusion-Derived Exosomal miR-130b-3p and miR-423-5p Promote the Proliferation of Lung Cancer Cells via Cyclin D1"

_ijms, 2024, doi:10.3390/ijms251810119_

Round 1

Reviewer 1 Report

Comments and Suggestions for Authors

1. The significance of the study is in contributing to the understanding of the relationship between tuberculosis (TB) and lung cancer, particularly how tuberculous pleural effusion (TPE)-derived exosomal miRNAs, specifically miR-130b-3p and miR-423-5p, promote the proliferation of lung cancer cells.
2. More detailed mechanistic studies should be applied to further elucidate how these miRNAs influence the Cyclin D1 signaling pathway.

3. Additional experiments could be conducted to explore whether these findings are consistent across different lung cancer cell lines or specific to A549 cells.

4. Make more comprehensive discussion of potential clinical applications, particularly how these findings could translate into therapeutic interventions for lung cancer patients with a history of TB.

Author Response

Reviewer 1 comments:

  1. The significance of the study is in contributing to the understanding of the relationship between tuberculosis (TB) and lung cancer, particularly how tuberculous pleural effusion (TPE)-derived exosomal miRNAs, specifically miR-130b-3p and miR-423-5p, promote the proliferation of lung cancer cells.

  1. More detailed mechanistic studies should be applied to further elucidate how these miRNAs influence the Cyclin D1 signaling pathway.

We agree with the reviewer's comments.

Among the various mechanisms regulating cyclin D1, this study focused solely on the NF-κB pathway. Therefore, we highlighted the need for further research into other mechanisms as a limitation of this study.

Reference:

Several oncogenic events can affect cyclin D1 abundance. Receptor tyrosine kinases (RTKs), the MEK-ERK, WNT as well as NF-kB pathways can lead to increased transcription of cyclin D1. Post-transcriptional effects on cyclin D1 include protein translation through the PI3K–mTOR pathway

MiR-130b and MiR-423 have been reported in previous studies to regulate these pathways. Monoe et at reported that miR-130 family upregulates a wide range of receptor tyrosine kinases (RTKs) signaling via the expression of phosphorylated Src. Li et al reported that miR-423-5p resulted in upregulation of important signaling molecules such as p-AKT and p-ERK1/2. 

References:

  1. Monoe Y, Jingushi K, Kawase A, Hirono T, Hirose R, Nakatsuji Y, et al. Pharmacological Inhibition of miR-130 Family Suppresses Bladder Tumor Growth by Targeting Various Oncogenic Pathways via PTPN1. Int J Mol Sci. 2021 Apr 29;22(9).
  2. Li S, Zeng A, Hu Q, Yan W, Liu Y, You Y. miR-423-5p contributes to a malignant phenotype and temozolomide chemoresistance in glioblastomas. Neuro Oncol. 2017 Jan;19(1):55-65.

“As a limitation, in addition to the NF-κB pathway demonstrated in this study, further mechanistic research should be conducted to explore whether these miRNAs also influence cyclin D1 by regulating other pathways.”

  1. Additional experiments could be conducted to explore whether these findings are consistent across different lung cancer cell lines or specific to A549 cells.

We conducted experiments using other human lung cancer cell lines in addition to the A549 cells, as suggested by the reviewer.

It was confirmed that, similar to A549 cells, an increase in phosphorylated P65 and cyclin D was induced by TPE-derived exosomes in SNU-2315 and SNU-1330 cells as well.

SNU-2315- human adenocarcinoma lung cancer

SNU-1330- human squamous cell carcinoma lung cancer

  1. Make more comprehensive discussion of potential clinical applications, particularly how these findings could translate into therapeutic interventions for lung cancer patients with a history of TB.

“From the perspective of personalized medicine, developing therapeutic targets or novel biomarkers for tuberculosis-related lung cancer requires a thorough understanding of the precise regulatory mechanisms of EV miRNA transfer and their biological functions to identify target genes. A key challenge in adapting miRNA inhibitors for therapy is increasing their delivery efficiency. Additionally, establishing a large-scale tuberculosis-related lung cancer cohort is essential for validating candidate miRNAs and target genes.”

Reviewer 2 Report

Comments and Suggestions for Authors

Kang et. al. demonstrated that tuberculous pleural effusion-derived exosomes (TPE exosomes) promote the proliferation and migration of lung cancers. Additionally, two miRNAs, miR-130b-3p and miR-423-5p, were identified as upregulated miRNAs in A549 cells treated with TPE exosomes and were shown to be key effectors for A549 tumor growth and migration. This study introduces an interesting result regarding the relationship between TB and lung cancer, but several concerns dampen the initial enthusiasm. Hope that the comments below will help improve the novelty, integrity, and logical flow of the manuscript.

1.       In this study, the authors used exosomes obtained from TPE or transudative PE. The terms might be organized for a better understanding of the materials such as exudative TPE exosomes v.s. transudative PE exosomes or TB exosomes v.s. non-TB exosomes.

2.       In result 2.1, Line 72, the description of PKH26 for cell membrane staining can be added.

3.       In Figure 1B, the expression of exosome markers was not in an identical range in the TPE-exosome group. For this study, TPE or transudative PE exosomes were obtained from two patients for each group. If there are differences in exosome marker expression according to the patient which can indicate different cargo patterns, they should be separately grouped and analyzed.

4.       Do the exudative and transudative PE have identical characteristics? And could the transudative PE be recognized as a suitable control exosome source against exudative TPE? If the authors used exudative non-TB PE exosomes, could other potential miRNA candidates be identified?

5.       The results for Figure 2A were not described. If Figure 2A is the miRNA profile of TPE exosomes compared to T exosomes, there are many distinguished miRNAs specified in TPE exosomes rather than miR-130b-3p and miR423-5p. Discussion about those distinguished miRNAs and their related target proteins and gene ontology would be important information to characterize TPE exosomes.

6.       Figures 2B and 2C showed that the TPE exosome-treated cells had increased levels of miR-130b-3p and miR-423-5p. But miR-320b which was also abundant in TPE exosomes was not increased in TPE exosomes-treated cells. Also, miR-372a-5p and miR-409-3p which were abundant in T exosomes were not increased in T-exosome-treated cells. Thus, it appears that cellular miRNA expression was changed by TPE or T exosome treatment but was not related to the amount of exosomal miRNA delivered by TPE or T exosomes. It might be an important discussion point.

7.       In Figures 3C and 4C, western blot images did not show obvious differences among the treatment groups. Image-based quantification of each band intensity might help exhibit the differences.

8.       In Figure 4, GW4869 was combined with TPE exosomes. However, it was not mentioned in the Discussion at all, so the readers would not properly understand the purpose of these experiment sets. Figure 4 indicates that blocking the exosome release from A549 cells treated with TPE-exosomes decreased cellular migration further. Thus, it might be emphasized that exosome-mediated crosstalk between TPE-exosome-treated A549 cells enhances the migration of A549 cells.

9.       In Abstract, Line 20, ‘select exosomal microRNAs’ may mean ‘the selected exosomal microRNAs’.

In Abstract, Line 23, ‘select microRNAs’ may mean ‘the selected microRNAs’.

Caption for Figure 4C was missing.

Author Response

Manuscript ID: ijms-3183776

Title: Tuberculous pleural effusion-derived exosomal miR-130-3p and miR-423-5p promote the proliferation of lung cancer cells via Cyclin D1

Dear Editor and reviewers,

On behalf of our co-authors, we are pleased to submit the revised manuscript entitled " Tuberculous pleural effusion-derived exosomal miR-130-3p and miR-423-5p promote the proliferation of lung cancer cells via Cyclin D1"(Manuscript ID: ijms-3183776) for reconsideration as an Original Article in International Journal of Molecular Sciences. 

We appreciate the careful and thoughtful comments and we have provided point-by-point responses below. We have also included the revised manuscript with Tracked Changes. 

Please get in touch with us with any questions or concerns.  Thanks very much for your assistance.

Sincerely,

Ji Young Hong, M.D., Ph.D.          

Reviewer 2 comments:

Kang et. al. demonstrated that tuberculous pleural effusion-derived exosomes (TPE exosomes) promote the proliferation and migration of lung cancers. Additionally, two miRNAs, miR-130b-3p and miR-423-5p, were identified as upregulated miRNAs in A549 cells treated with TPE exosomes and were shown to be key effectors for A549 tumor growth and migration. This study introduces an interesting result regarding the relationship between TB and lung cancer, but several concerns dampen the initial enthusiasm. Hope that the comments below will help improve the novelty, integrity, and logical flow of the manuscript.

  1. In this study, the authors used exosomes obtained from TPE or transudative PE. The terms might be organized for a better understanding of the materials such as exudative TPE exosomes v.s. transudative PE exosomes or TB exosomes v.s. non-TB exosomes.

We compared exudative tuberculous pleural effusion-derived exosomes (TPE-exosomes) with transudative pleural effusion-derived exosomes (T-exosomes).

A representative exudative pleural effusion includes malignant and infectious effusions, and it has been reported that distinct genes and biological processes are differentially expressed between the two groups. Our study specifically focuses on tuberculous pleural effusion among infectious effusions. In this study, we used transudative PE exosomes as a reference control group like the previous paper.

<References>

Zamora-Molina L, Garcia-Pachon E, Amoros M, Gijon-Martinez J, Sanchez-Almendro J, Baeza-Martinez C, et al. Transcriptomic Profiling of Pleural Effusions: Differences in Malignant and Infectious Fluids. Medicina (Kaunas). 2024 Mar 1;60(3).

“The transudative PEs were used as a reference control group. The clinical characteristics of the patients are listed in Table 1. We compared exudative tuberculous pleural effusion-derived exosomes (TPE-exosomes) with transudative pleural effusion-derived exosomes (T-exosomes).”

  1. In result 2.1, Line 72, the description of PKH26 for cell membrane staining can be added.

We have revised the manuscript according to the reviewer’s suggestions.

“PKH26 was used as a cell tracer by binding to the cell membrane of A549 cells. A549 cells and TPE-derived exosomes were stained with PKH26 (red) and PKH67 (green) dyes, respectively. Thus, PKH67-labeled TPE-derived exosomes were incubated with PKH26 labeled A549 cells for 3 h, after which the fluorescence intensity was measured using flow cytometry.”

  1. In Figure 1B, the expression of exosome markers was not in an identical range in the TPE-exosome group. For this study, TPE or transudative PE exosomes were obtained from two patients for each group. If there are differences in exosome marker expression according to the patient which can indicate different cargo patterns, they should be separately grouped and analyzed.

Although there is some variation in exosome marker intensity within the TPE-exosome group, it is clear that exosome markers are increased compared to the A549 cells (negative control). To clearly demonstrate the impact on lung cancer for each individual case, we have presented the invasion assay and western blot as individual data sets.

  1. Do the exudative and transudative PE have identical characteristics? And could the transudative PE be recognized as a suitable control exosome source against exudative TPE? If the authors used exudative non-TB PE exosomes, could other potential miRNA candidates be identified?

In previous literature, transudative pleural effusion (PE) has commonly been used as a reference control group. Exudates and transudates differ in their underlying mechanisms and exhibit distinct characteristics in the nature of the effusion, which are clinically distinguished using Light's criteria. Unfortunately, this study did not include a comparative analysis with exosomes extracted from other non-TB exudative pleural effusions, such as malignant pleural effusion or bacterial empyema. Given previous studies that have demonstrated differences in gene expression or miRNA profiles between malignant pleural effusion and tuberculous pleural effusion among exudates, it is anticipated that the miRNA candidates of TPE-derived exosomes might differ if the comparison group were changed from transudative PE to non-TB exudative PE.

Reference

  1. Zamora-Molina L, Garcia-Pachon E, Amoros M, Gijon-Martinez J, Sanchez-Almendro J, Baeza-Martinez C, et al. Transcriptomic Profiling of Pleural Effusions: Differences in Malignant and Infectious Fluids. Medicina (Kaunas). 2024 Mar 1;60(3).
  2. Gautam S, K CS, Bhattarai B, K CG, Adhikari G, Gyawali P, et al. Diagnostic value of pleural cholesterol in differentiating exudative and transudative pleural effusion. Ann Med Surg (Lond). 2022 Oct;82:104479.
  3. Zhang X, Bao L, Yu G, Wang H. Exosomal miRNA-profiling of pleural effusion in lung adenocarcinoma and tuberculosis. Front Surg. 2022;9:1050242.

  1. The results for Figure 2A were not described. If Figure 2A is the miRNA profile of TPE exosomes compared to T exosomes, there are many distinguished miRNAs specified in TPE exosomes rather than miR-130b-3p and miR423-5p. Discussion about those distinguished miRNAs and their related target proteins and gene ontology would be important information to characterize TPE exosomes.

Upregulated miRNA

Target gene

Downregulated miRNA

Target gene

miR-130b-3p

miR-423-5p

miR-142-5p

miR-107

miR-320b

miR-185-5p

PLAG1 BACE1 MYB VEGFA HIF1A ARNT FBXW7 GRN DAPK1 KLF4 CYP2C8 CHRM1 NOTCH2 AGO1 PRKCE FOXO1 CDK8 CAV1 LATS2 JAK1 IL6 RAD51 SALL4 SH3GL2 SERPINB5 CHGA CLOCK LIN28A NF1 CDC42 HMGA2 CPEB1 SIAH1 HOXA1 PDGFRA ITGB1 CMPK1 FMR1 PPARGC1A DLL1 NKD2 CYLD IGF1 CYP2C9 CCDC6 PPARA UCP1 SCD SNAI3 MST1 SAV1 FOS CXCR4 JUN NR5A2 NOTCH1 ROCK2 FAM162A RHOT1 KRAS MYO6 DNMT3A FNDC3B FHIT MACC1 PTGS2 JAG1 MDM2 SDC1 RREB1 KLF5 EDNRA DIO1 PEBP1 TCEAL1 PHLPP1 DPYSL2 HOXD10 PTX3 MBD2 SERPINF2 TRIB1 PAK2 CASP7 MTOR PHLPP2 RASSF8 LAMP1 GJA1 ST14 TRAPPC2B PPARG CYP3A4 PAX3 CCNT1 PAX7 SEMA6A VEGFC CREB1 ABCA1 PSAP MFF DPYD SHC1 THBS2 THBS1

miR-374a-5p

miR-409-3p

miR-125b-5p

miR-224-5p

miR 139-5p

GLI1 NKIRAS2 SMO TP53 VDR BAK1 ERBB3 ERBB2 BMF NTRK3 AKT1 CYP24A1 RAF1 PRDM1 GRIN2A CDKN2A E2F3 IGF2 LIN28B BBC3 PPP1CA PRKRA ETS1 RPS6KA1 TNFAIP3 PIGF BCL3 TBC1D1 DGAT1 FGFR2 ARID3B SMAD4 MCL1 IL6R STARD13 ABTB1 HK2 E2F2 MMP13 MAPK14 EPO MUC1 NES CDH5 ARID3A BCL2L2 EIF5A2 MXD1 PIAS3 PIK3CD LIPA IKZF2 IKZF3 IKZF4 ICAM2 VPS4B SET CCNJ ENPEP MEGF9 MAN1B1 EPOR AHRR SCNN1A VPS51 SIRT7 DUSP6 TET2 SPHK1 MMP2 MMP26 MAP3K11 SFRP5 GAB2 Fas APLN PIK3CB IGF1R NEU1 ALOX5 CD44 EGFR TNF PHF8 KLC2 ANGPT2 APC FES DKK3 FZD6 CDKN2D PODXL JAK2 SUV39H1 PTH1R HOTTIP DRAM2 TGM2 FOXA1 TP53INP1 CSF1 ZHX2 PIK3CA NFKB1 HRAS OIP5 ACTC1 ADGRL4 SMARCA4 TPD52 MET BCL2 PDE4D WNT1 MMP11 NRIP1 CD38 ATP6AP2 ITGA6 MARCKSL1 COL4A1 GPC1 FN1 ATP8A1 RET PHB CCNA2 HIP1R PLK2 ROR1 CCNG1 FZD7 OSBPL6 HMGB3 CDH11 CX3CL1 UCA1 PINK1 CDK6 SLC7A6 TMEM2 CYR61 VEZT XBP1 DLX5 CCND1 PAX6 CDC25A ACVR1 MAX DICER1 ATM GADD45A SRCIN1 WNT5A WIF1

In terms of KEGG pathway related to miRNA, TPE exosomes show a significantly higher association with cancer pathways compared to T exosomes. In terms of gene ontology related to miRNA, TPE exosomes show a significantly higher association with epithelial cell proliferation, cell to cell junction, transcription regulator complex, DNA binding transcription activator compared to T exosomes.

However, to confirm the directional expression of individual genes targeted by miRNAs in each group and their direct regulatory mechanisms, in vitro experiments are required.

  1. Figures 2B and 2C showed that the TPE exosome-treated cells had increased levels of miR-130b-3p and miR-423-5p. But miR-320b which was also abundant in TPE exosomes was not increased in TPE exosomes-treated cells. Also, miR-372a-5p and miR-409-3p which were abundant in T exosomes were not increased in T-exosome-treated cells. Thus, it appears that cellular miRNA expression was changed by TPE or T exosome treatment but was not related to the amount of exosomal miRNA delivered by TPE or T exosomes. It might be an important discussion point.

After exosomes are delivered to recipient cells, some of the miRNAs they contain function within the cell, while others undergo degradation. This difference is determined by factors such as miRNA stability, the intracellular environment, interactions with target mRNAs, and the metabolic state of the cell.

“Exosomes mediate their effects on recipient cells through a multi-step process: interaction with cell surface receptors, internalization via endocytosis, and the release of the endo-some’s contents into the cytoplasm, where they carry out their functional effects [27]. After exosomes are delivered to recipient cells, some of the miRNAs they contain function within the cell, while others undergo degradation. This difference is determined by factors such as miRNA stability, the intracellular environment, interactions with target mRNAs, and the metabolic state of the cell. The delivery of specific miRNAs reflects the interplay between exosomal surface proteins and ligands, receptor specificity, environmental factors, host genetics, and the structural characteristics and stability of the miRNA [28, 29].”

  1. In Figures 3C and 4C, western blot images did not show obvious differences among the treatment groups. Image-based quantification of each band intensity might help exhibit the differences.

As suggested by the reviewer, image-based quantification of each band intensity was performed.

  1. In Figure 4, GW4869 was combined with TPE exosomes. However, it was not mentioned in the Discussion at all, so the readers would not properly understand the purpose of these experiment sets. Figure 4 indicates that blocking the exosome release from A549 cells treated with TPE-exosomes decreased cellular migration further. Thus, it might be emphasized that exosome-mediated crosstalk between TPE-exosome-treated A549 cells enhances the migration of A549 cells.

“Our data from both in vivo and in vitro studies confirmed that TPE-derived exosomes promote lung cancer proliferation, and this effect was reduced when GW4860, an inhibitor of exosome biogenesis, was administered. We focused on the miRNA component of the exosomes.”

  1. In Abstract, Line 20, ‘select exosomal microRNAs’ may mean ‘the selected exosomal microRNAs’.

We have made the changes as per the reviewer's suggestion.

In Abstract, Line 23, ‘select microRNAs’ may mean ‘the selected microRNAs’.

We have made the changes as per the reviewer's suggestion.

Caption for Figure 4C was missing.

We have added a legend for Figure 4C.

“(c) Expression of p65 and G1/S regulatory molecules as measured via Western blotting in A549 cells treated with TPE-derived exosomes with or without inhibitors of miR-130b-3p and miR-423-5p. All experiments were performed in triplicate. The results are expressed as the mean ± SEM of triplicates.”

Round 2

Reviewer 1 Report

Comments and Suggestions for Authors

no more recommendation

Reviewer 2 Report

Comments and Suggestions for Authors

The reviewer's comments were addressed properly.